# Perceptions, Representations, and Experiences of Patients Presenting Nonspecific Symptoms in the Context of Suspected Lyme Borreliosis

**DOI:** 10.3390/microorganisms9071515

**Published:** 2021-07-15

**Authors:** Alice Raffetin, Aude Barquin, Steve Nguala, Giulia Paoletti, Christian Rabaud, Olivier Chassany, Pauline Caraux-Paz, Sarah Covasso, Henri Partouche

**Affiliations:** 1Department of Infectious Diseases, Tick-Borne Diseases Reference Centre, North Region, CH Villeneuve Saint Georges, 40 Allée de la Source, 94190 Villeneuve-Saint-Georges, France; steve.nguala@chiv.fr (S.N.); pauline.caraux-paz@chiv.fr (P.C.-P.); sarah2597@hotmail.fr (S.C.); 2European Study Group for Lyme Borreliosis ESGBOR, ESCMID, Gerbergasse 14 3rd Floor, 4001 Basel, Switzerland; 3Département de Médecine Générale, Paris University, Site Cochin 27, Rue du Fbg Saint-Jacques, CEDEX 14, 75679 Paris, France; audebarquin@live.fr (A.B.); henri.partouche@parisdescartes.fr (H.P.); 4Department of Psychiatry, Tick-Borne Diseases Reference Centre, Île-de-France/Hauts-de-France, CH Villeneuve Saint Georges, 40 Allée de la Source, 94190 Villeneuve-Saint-Georges, France; giulia.paoletti@gmail.com; 5Department of Infectious Diseases, Tick-Borne Diseases Reference Centre, East Region, CHRU Nancy, Bâtiment Philippe Canton, Hôpitaux de Brabois, Allée du Morvan, 54500 Vandoeuvre les Nancy, France; c.rabaud@chru-nancy.fr; 6Health Economics Clinical Trial Unit (URC-ECO), Hôpital Hotel-Dieu, AP-HP, 1 Place du Parvis Notre Dame, 75004 Paris, France; olivier.chassany@aphp.fr; 7Department of Anthropology, University Lyon II, UFR Anthropologie, Sociologie et Science Politique, Université Lumière Lyon 2, 5 Avenue Pierre Mendès France, 69676 Bron, France

**Keywords:** Lyme borreliosis, perceptions, representations, experiences, qualitative study, grounded theory

## Abstract

Background: Some subjective symptoms may be reported at all stages of Lyme borreliosis (LB) and may persist for several months after treatment. Nonspecific symptoms without any objective manifestation of LB are sometimes attributed by patients to a possible tick bite. The aim of our study was to explore the perceptions, representations, and experiences that these patients had of their disease and care paths. Methods: We performed a qualitative study through individual interviews (October 2017–May 2018), based on grounded theory, following the COREQ checklist. A balanced sample of patients with diverse profiles was recruited at consultations with general practitioners and infectious disease physicians. Results: Twelve patients were interviewed. Data saturation was reached at the twelfth interview. For codes, 293 were identified, and classified into 5 themes: (1) the experience of disabling nonspecific symptoms, especially pain, causing confusion and fear, (2) long and difficult care paths for the majority of the patients, experienced as an obstacle course, (3) a break with the previous state of health, causing a negative impact on every sphere of the patient’s life, (4) empowerment of the patients and the self-management of their disease, and (5) the strong expression of a desire for change, with better listening, greater recognition of the symptoms, and simpler care paths. Conclusions: This study allows for the understanding of a patient’s behaviours and the obstacles encountered, the way they are perceived, and the necessary solutions. The patients’ expectations identified here could help physicians better understand the doctor–patient relationship in these complex management situations, which would reduce the burden of the disease. The current development of specialised reference centres could help meet the patients’ demands and those of family physicians.

## 1. Introduction

Lyme borreliosis (LB) is the most reported tick-borne infection in North America and Europe [1,2]. In France, the incidence rate was estimated at 104 per 100,000 inhabitants in 2018 [3]. The increase in LB cases has been attributed to the better diagnosis, better notification of LB cases, and increasing global temperatures. Climate change seems to influence vector–host–pathogen interactions [4,5].

LB is a tick-borne bacterial infection caused by the spirochaete *Borrelia burgdoferi* sensu lato, transmitted by *Ixodes* tick bites [6]. A tick can also be the vector of other pathogens [7,8]. *Borrelia burgdorferi* rarely occurs with other co-infections [9,10].

LB is a multisystemic disease, mainly affecting the skin, nervous system, and joints. LB evolves in three clinical stages: early localised (erythema migrans), early disseminated (evolution < 6 months), and late disseminated stage (evolution > 6 months) [7,11].

According to French, European, and American guidelines, the diagnosis of a confirmed LB is based on tick exposure, specific clinical manifestations, and a positive serological test according to a two-tiered approach (initial ELISA test followed by a Western blot confirmation test) for the disseminated stage [8,11,12,13,14].

Antibiotic therapy is recommended for all the stages of the disease, with different molecules and durations according to the stage of the disease, and not exceeding 28 days [8,13,14]. Five blind randomised clinical trials found no evidence that prolonged antibiotic treatment was beneficial; on the other hand, severe adverse events were observed [15,16,17,18,19].

Some subjective symptoms (asthenia, polyalgia, memory or concentration complaint, etc.) may exist at all of the stages and may persist after a well-conducted antibiotic therapy. This is called the post-treatment Lyme disease syndrome (PTLDS). The diagnostic challenge is that these symptoms may be associated with LB and/or with different diagnoses [20,21]. The hypothesis of a LB is rejected based on a set of arguments: absence of tick-exposure just before the onset of the symptoms, non-evocative clinical manifestations, negative serological test, antibiotic failure, and another disease corresponds better to the clinical picture. This setting could complicate patient management in clinical practice. Jeoffrion et al. compared the representations of the disease by health professionals and non-professionals [22]. The authors observed that the social perceptions of the disease differed between the groups, depending on the amount of personal experiences for the non-professionals and the characteristics and concrete involvement with the disease for the professionals [22]. These findings reveal the importance of knowing and accounting for these representations to promote understanding and trust between healthcare professionals and non-professionals [23,24].

One American study and two French studies, which included patients with persistent symptoms linked to LB, reported that these patients experienced LB as a burden with major social repercussions and impact on their daily life (chronic pain, fatigue, etc.) [25,26,27]. The study of Drew and Hewitt found similar results [28].

The aim of our study is to highlight the perceptions, representations, and experiences of the patients presenting persistent symptoms and suspected LB.

## 2. Materials and Methods

We used a qualitative method that aims to understand social phenomena in their natural environment, including the points of view of the participants [23]. This holistic vision provides answers when investigating poorly documented data, subjective factors that are difficult to measure, and representations and perceptions, in which the applications are concrete in a healthcare relationship [24].

A balanced sample was sought to maximise the variation in opinions and experiences. The sample size was not initially estimated, in accordance with good practices in qualitative research [29,30]. The final interview was dictated by the saturation of the data (no new information provided in a subsequent interview).

The eligibility criteria were the following. The patients had to be over 18 years of age, to be living in metropolitan France (in both endemic and non-endemic areas), and to have or to have had a suspicion of LB with persistent symptoms on-going for at least six months, with or without a diagnosis of LB, treated or not. The sample included individuals who believed they suffered from LB but whose doctor did not share this opinion.

Patients were invited to participate in a face-to-face setting during a general medical consultation or an infectious disease consultation. Not all the patients recruited had a complaint related to a suspicion of LB as a reason for the consultation, but this suspicion was mentioned during the medical visit, either while discussing the patient’s medical history or in some other way.

After the patients consented to participate, audio-recorded semi-structured individual interviews took place face-to-face or by telephone (A.B.) between July 2017 and March 2018. The interviews allowed for free expression, which was particularly suited for identifying participants’ perceptions and experiences. We used the comprehensive interview method of Kaufmann [31].

The interview guide had been developed earlier based on bibliographic data and the objectives of our study. We had to first deconstruct our preconceived ideas concerning the patients’ movement between clinics in search of an appropriate diagnosis and treatment (difficulty in receiving a diagnosis of late LB, a lack of research into differential diagnoses, and a lack of responses adapted to their symptoms), their feelings of stigmatisation, their lack of confidence in the French healthcare system and its doctors, and their fatalism about their LB, which they felt could not be cured. The interview guide consisted of an introductory part and eight open-ended questions: (1) What are the reasons your symptoms best match LB? (2) How do you feel about this disease, and how did you feel when you were diagnosed? (3) How has the disease affected your life? (Daily impact) (4) How do you explain this disease? (5) How do you think you can be cured of this disease? Or how did you recover from it? (6) Which treatment has helped you the most or best met your needs and expectations? (7) What were the difficulties and obstacles you encountered during your care path? (8) If you were the Minister of Health, what would you do to improve the diagnostic and therapeutic management of LB? Finally, a satisfaction question was asked at the end of the interview. The interview guide evolved as the interviews progressed so as to compare the opinions collected and to respect the semi-structured format, allowing new ideas to emerge from the participants. We asked patients to disregard the interviewers’ medical status (A.B.) to avoid confusion with a medical consultation. A.B. performed the interviews with an anthropological approach and not a medical approach. This point was clarified at the beginning of each interview with the patient. Moreover, A.B. was not involved in the care of patients and did not belong to the care structure where the patients were recruited.

Interviews were transcribed in writing immediately after the conclusion of each interview. A.B., A.R., and S.C. (anthropologist) manually carried out blind open-ended coding and thematic analysis. We followed an inductive approach, allowing us to formulate hypotheses as we collected data [23,32]. This approach falls within the framework of grounded theory, where the data collected are coded and then grouped by theme in an iterative process of progressive theorisation of a phenomenon, characterised by the simultaneous collection and analysis of data [29]. The items of the COREQ grid (consolidated criteria for reporting qualitative studies) were followed in this work [30].

The protocol was approved by the ethics committee of CPP Île-de-France VII, number 17-054. Consent was obtained from each participant. The study is registered with ANSM (2017-A01709-44) and CNIL (X902218725D).

## 3. Results

Thirteen patients were approached, one of whom declined due to a lack of interest. The sociodemographic and clinical characteristics of the 12 patients interviewed are summarised in Table 1.

The sex ratio was 1, and the mean age was 53.2 years (30 to 72). Two patients were members of an association of activist patients, two wished to join, and one had contacted an association for information. During the interviews, four clinical situations emerged: (1) five patients with a confirmed diagnosis of LB, treated according to the recommendations, with persistent symptoms; (2) three patients with a diagnosis of LB disproved, who had received or were receiving treatments contrary to recommendations, with persistent symptoms; (3) three patients undergoing diagnostic evaluation for unexplained symptoms, with no obvious diagnosis; (4) one patient with a confirmed diagnosis of LB, treated according to the recommendations, without persistent symptoms. The average duration of the interviews was 38 min (15 to 50 min). Three interviews were conducted face-to-face, and nine by telephone. The interviews generated 293 codes, which were grouped into 5 non-predefined themes, discussed in detail below. Data saturation was reached at the twelfth patient included.

### 3.1. Theme 1—A Painful Experience with the Disease, Leading to Confusion and Fear

#### 3.1.1. Non-Specific Symptoms, Sometimes Severe, Dominated by Pain and Asthenia


*P8: ‘Nothing could bring me relief (...), the pain was almost unbearable’. P4: ‘[A]ways tired, tired (...) tired, tired’.*


#### 3.1.2. Incomprehension, Fear, and Doubt when Faced with the Lack of Explanation for the Symptoms


*P3: ‘[W]e kept doing the analyses, we didn’t understand.’ P5: ‘It’s the chameleon symptom again’. P9: ‘Several times I said to myself: “But my poor girl, but it’s you who is doing this to you!”’*


As soon as the LB hypothesis was mentioned once in the course of their illness, the patients linked all their symptoms to it. *P1: ‘When you list all the symptoms of Lyme, it’s (hesitation), it’s obvious.’* The patients expressed a feeling of fear of unpredictable flare-ups, of not being cured, etc., *P7: ‘I was afraid of not knowing how I would end up’.*

### 3.2. Theme 2—A Long and Difficult Treatment Path, Experienced as an Obstacle Course

#### 3.2.1. A Fight against a Vicious Disease, Caused by Super-Intelligent and Invisible Bacteria (P3): A Fight Lost in Advance?

*P5: ‘It’s impossible to destroy because they hide, and they proliferate and you have to keep attacking’.* LB was seen as inevitable; sometimes, as necessary. *P3: ‘I tell myself that it’s that I must get sick and that is going to bring me something’.*

#### 3.2.2. A Fight against the Medical World?

The lack of explanation was attributed to a lack of knowledge, interest, or consideration from the physicians, and seen as a barrier to diagnosis. *P5: ‘My GP, I am reluctant to ask him, he doesn’t want to believe me’. P1: ‘They don’t listen, (...) they look at everything medical, and as long as the tests are negative, they say that you have nothing’.* The absence of consensus on recommendations at the time of the study has reinforced the feeling of abandonment by the scientific community.

#### 3.2.3. A Fight against the Healthcare System? From Medical Nomadism to the Misuse of the Healthcare System and the Search for Alternative Systems

*P3: ‘I went for multiple tests for months and months that didn’t find anything. I think I must have been to five hospitals (hesitation), about fifteen doctors’.* Some patients turned to more attentive ’hidden’ (P3) doctors, at high cost, with uncertain results. *P1: ‘Here I am at 22,000 euros for a treatment in Germany (...). It has already given me an answer, but it’s not enough to have an answer, the body must also be healed’*.

#### 3.2.4. The End of the Fight Is in Sight? A Trusted Doctor, a Specific Diagnosis, and a Coordinated Care Pathway

After a long struggle, several patients met competent physicians, who provided answers, within the regular healthcare system. Younger physicians seemed more attentive. Coordination of the care path by a doctor, whether a general practitioner (GP) or an infectious disease specialist, seemed to be crucial to the treatment process. *P7: ‘I only go to Dr X. now’*. The announcement of the diagnosis put an end to the doubt. *P7: ‘I was free, we had found what was going to bring me back to my life’.*

### 3.3. Theme 3—A Negative Impact on All Areas of the Patient’s Life

#### 3.3.1. Disease Taking a Serious Toll on the Patient’s Health


*P7: ‘When you are hyperactive like me and you suddenly see yourself diminished on a couch like that’.*


#### 3.3.2. Multiple and Negative Repercussions, Experienced as an Injustice

At the professional level, the patients reported absences linked to multiple medical consultations, repeated leave from work, etc. Activities were impacted by the unpredictability of the symptoms, leading to the feeling of being overwhelmed by the disease. The patients described either a lack of understanding from their relatives, or unconditional support, sometimes with the family adapting to their condition. *P1: ‘[W]e made an extension downstairs, because I could no longer climb the stairs (...) I have my husband who has always been there (emotional voice)’.*

### 3.4. Theme 4—Patients’ Empowerment

#### 3.4.1. Patients Organising a Network to Discuss Their Experiences and Feel Understood

*P2: ‘There is contact with other people who have been, if you understand, affected, and who understand the disease’*.

#### 3.4.2. Self-Education: Patients who Educate Themselves about the Illness and Inform Others

Some patients were informing themselves, until they felt they were experts. *P3: ‘[W]e read a lot of books, and from there we understood what it was, and so we were able to identify it’.* There was mistrust of information on the Internet, in contrast with information from associations or scientific articles. Often the diagnosis had been mentioned by a knowledgeable relative, or by the patient himself or herself.

#### 3.4.3. Self-Medication and the Patient-Therapist: Patients Experimenting on Themselves?

Some patients self-medicated, others *‘[begged]’* (P5) their doctor for antibiotics, and others consulted *‘hidden’* doctors prescribing various long-term antibiotics treatments. Those seeking relief described the lack of alternatives, which had *‘[forced]’ (P3)* them to continue for fear of recurrence. Others preferred non-drug approaches. *P3: ‘[T]he Lyme specialist (...) gave me three months of antibiotics, and there I almost lost my life once a day (...) we went all the way, but in the end we said no, we’ll see about something natural’.*

#### 3.4.4. Theorisation of the Illness


*P3: ‘You don’t necessarily get it from tick bites; it can be passed on from generation to generation’.*


### 3.5. Theme 5—A Desire for Change

#### 3.5.1. Need for Listening and Recognition

The patients accepted that the diagnosis could be long and difficult, but they did not accept the fact that no one listened to them, which was their main barrier to care. The experience of being listened to attentively by a doctor was the main helping factor. *P9: ‘[T]he doctor was great, because she actually listened to me, (...) she even heard me’*.

#### 3.5.2. Improved Knowledge of the Disease and the Diagnostic Tests, with the Patients Involved in the Research


*P5: “I am always happy if we can spread the knowledge of this disease, about which so little is known’.*


#### 3.5.3. Raising Physicians’ Awareness, Training Specialists, and Launching Specialised Services for a Better Care Pathway


*P5: ‘At least the doctors are informed and trained, because that’s what’s incredible’. P12: ‘I think what is good about hospital X is that there is a team dedicated to this Lyme disease, they are competent people’.*


#### 3.5.4. From the Patient-Activist to Not Understanding the Controversy

Some patients said they were tired of the polemics surrounding LB, which they perceived as a barrier to increased knowledge. *P7: ‘I trust the doctor completely, not the quacks’. P11: ‘[You] go on the internet, type in “Lyme disease”, and it’s (hiss) help! (laughter)’.*

## 4. Discussion

### 4.1. Modelisation of the Theory

The thematic analysis of the verbatim reports allowed a model to be developed that considers the perceptions, representations, and experiences of patients with persistent symptoms in the context of suspected LB (Figure 1).

The illness caused multiple feelings perceived by the patients: incomprehension, fear, isolation, guilt, disappointment, anger, injustice, and doom. The patients had a representation of a vicious and dangerous disease, responsible for a rupture with the previous state of health. The experience of the illness was painfully experienced with the problem of non-specific, unpredictable, and severe symptoms. Patients reported difficulty for physicians in interpreting their symptoms and a lack of listening and recognition, responsible for a difficult care path. The diagnostic announcement was viewed as a recognition of the reality of their symptoms. This experience of illness, of the physician–patient relationship, and of the care path was the driver of the empowerment of the patients: self-management of their disease, re-organisation of their way of life, and involvement in the fight against their disease. Patients wished for changes such as better and attentive listening, recognition, better medical and societal management, and improved knowledge on Lyme borreliosis and persistent symptoms.

### 4.2. Points of Discussion Highlighted by This Study

#### 4.2.1. A Destructive Disease, a Liberating Disease, and a ‘Professional’ Disease

Herzlich proposes three types of representation of the disease: destructive, liberating, and ‘professional’ [33]:
With a destructive disease, the patient sees the abandonment of his or her social role as being excluded and adopts a passive attitude towards care.With a liberating disease, the inactivity generated by the disease frees the patient from his or her burdens, allowing him or her to carry out activities that he or she had not had the time to do previously. The disease is accepted.With a ‘professional’ disease, the patient accepts the mission of fighting the disease. Inactivity becomes acceptable, freeing the patient to actively combat the disease.

In our research, all three elements of the representation of the disease were present.

#### 4.2.2. Information Exchange at the Core of the Doctor–Patient Partnership

In France, the patient has an active role in his or her care, actively using his or her knowledge, perceptions, representations, and experiences. The doctor–patient partnership is based on making a shared medical decision [34]. According to a survey, one-third of general practitioners experience difficulty when faced with the ‘insistent’ demands of ‘hyper-informed’ patients [35]. Our study highlights that some physicians may also experience a lack of knowledge and information about LB, increasing the difficulty to answer the patient’s needs. The major challenge for the doctor is to determine on one hand the limits of his own knowledge and his capacity to answer the patient, and on the other hand the quality of the patient’s information sources. Open dialogue, verification of sources, and data analysis are fundamental to making an informed shared medical decision.

#### 4.2.3. Improved Physician Education and Updated Scientific Guidelines

The patients highlighted the poor training of physicians regarding persistent symptoms, as has also been shown in several studies on somatic symptom disorders [36,37,38]. These results are consistent with the views of the GPs interviewed in study of Lisowski et al., 87% of whom said they were uncomfortable following up with patients who had symptoms after a full course of antibiotics due to having failed to provide codified management [35]. Nevertheless, continuing education on LB has increased, enabling better diagnosis, as evidenced by the increase in the number of diagnosed cases (3). Patients we interviewed, who reflected the associations, requested new guidelines for LB management. An update in the recommendations in several European countries shows that the scientific community is listening and wants to improve the management of the illness [8,12,13,20,39,40,41,42]. Post-treatment Lyme disease syndrome has been recognised [7,13,20]. France has taken a multidisciplinary approach to care, with a focus on listening to patients and recognising their symptoms [7,8,13].

#### 4.2.4. The Creation of Specialised Facilities: Putting an End to the Obstacle Course?

In 2019, five reference centres for tick-borne diseases, as well as associated centres, had been appointed by the Ministry of Health in France. Patients that we interviewed in this study who had received care from specialised facilities were satisfied. Following a dedicated one-hour consultation that included listening, recognition of the symptoms, and a discussion of broad management strategies, these multidisciplinary facilities propose a personalised care plan, adapted to the patient’s needs, whether the patient presented LB or a differential diagnosis. All patients with suspected LB are eligible to attend at the request of their physician. The challenge is to make these new organisations better known and ensure they are understood by the GPs, who have an essential role in the early management of complex symptoms. The main expectation that GPs have for these centres is diagnostic assistance in complex cases to avoid misdiagnosis [43]. It is interesting to note that these expectations are identical to those of the patients. Moreover, this raises the more general issue of the management of patients presenting long-lasting symptoms with significant repercussions in their daily life. These multidisciplinary approaches could be a successful model for many diseases and many patients.

### 4.3. Limitations and Strengths of the Study

Most patients (9/12) were enrolled in hospital consultations, which means that there was a risk that only severe cases would be included. However, this specialised consultation program was known to general practitioners in the region, who referred patients to it early, and eight patients of this study did not have severe symptoms. This point is therefore qualified. Of note, 6/12 patients have had a confirmed LB in the past, and 5/6 still had persistent symptoms, possibly related to PTLDS. Nonetheless, this point is balanced as 3/5 patients presented persistent symptoms for more than 5 years, probably suggesting a differential diagnosis and not only a PTLDS. Still, the participants, all of whom were motivated to actively improve the management of LB, might not represent all patients. The setting of the study is in France and may not correspond to other endemic countries for LB, because of cultural differences but also different healthcare systems. Nonetheless, this statement is balanced by the similar results obtained in other studies from other countries such as Canada and the USA [25,28,44,45]. Moreover, saturation was established at the twelfth interview without confirmation by a thirteenth.

To our knowledge, there are few other works on this topic. Four studies reported similar results regarding the long and difficult care pathway and the impact of the disease, giving internal validity to our study [25,26,27,45]. This makes the lack of confirmation permissible. The number of subjects included in the earlier studies was similar to that in ours.

The quality criteria of the international COREQ grid were observed. The interviews were carried out by a resident in general medicine (A.B.), who had never carried out an interview before. Follow-up and reformulation techniques were improved as the interviews proceeded. The interviewer did not know any of the participants. The interviewer’s medical status may have limited patients in their criticism of medical doctors and the healthcare system. This point was qualified as A.B. performed the interviews with an anthropological approach and not a medical approach. This point was clarified at the beginning of each interview with the patient. The varied profiles of the 12 patients allowed for the development of many perspectives.

Manual coding of the interviews and theorising were performed blindly by A.B., A.R., and S.C. Pooling the results showed overlaps in coding of the interviews and the theorisation presented above. The triangulation of the researchers is essential for validating the results.

One patient reviewed the transcript of his interview and the findings of the study, without correction.

We deconstructed our preconceptions before starting the study to avoid influencing the participants. Diagnostic and treatment wandering, and patient stigmatisation, were verified by our results. Patients’ low confidence in the French healthcare system was offset by their appreciation of their coordinating physician. The feeling of fatalism linked to LB was offset by the fact that some patients believed that they could be cured if the system changed.

## 5. Conclusions

The analysis of perceptions, representations, and experiences allows for the understanding of a patient’s behaviours and provides information on the obstacles encountered, the way they are perceived, and the necessary solutions. This makes it possible to improve patient care in essential ways. The results of our research meet this objective. The importance of a coordinated care pathway was evident, as was the need for careful listening and recognition. The fact that the importance of listening and recognising patients’ symptoms has been mentioned in several guidelines shows that the medical community is trying to meet these requirements [13,34]. Emphasising patients’ expectations could help physicians better understand the doctor–patient partnership in these complex diagnostic and management situations. The current development of specialised reference centres could help meet not only the patients’ demands with longer consultation and a multidisciplinary approach enabling attentive care, but also those of family physicians, who are at the heart of the healthcare system and responsible for its coordination.

The results of our study could be used to create a self-assessment questionnaire on the state of a patient’s health and the impact of the disease, specific to suspected LB patients, which could be used in the initial assessment, the follow-up, and the evaluation of therapeutic measures. The use of a validated questionnaire could help improve doctor–patient communication and systematically identify difficulties that patients may not have mentioned [46,47].

It would also be beneficial to conduct a similar research study a few years after the establishment of the new reference centres for suspected LB in order to assess their effectiveness with patients.

## Figures and Tables

**Figure 1 microorganisms-09-01515-f001:**
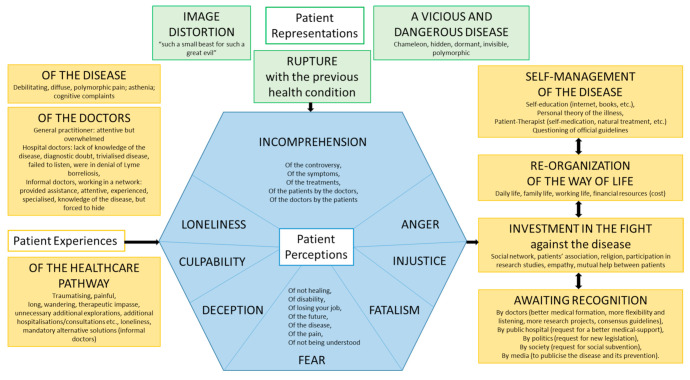
Model of the theory: perceptions, representations, and experiences of patients with persistent symptoms in the context of suspected Lyme borreliosis.

**Table 1 microorganisms-09-01515-t001:** Characteristics of the patients.

	Sex	Age (Years)	Marital Status	Profession	Status	Place of Residence	Recruitment	Duration of Symptoms	History of Tick Bite	History of EM	Clinical Situation *
P1	F	52	In couple	Accountant	Invalidity	Semi-rural	IDP	8.5 years	No	No	2
P2	M	72	In couple	Police officer	Retired	Rural	GP	5 years	Yes	Yes	1
P3	F	32	Single	Employee in a school	Work stopping	Rural	IDP	2 years	No	No	2
P4	F	64	In couple	Housewife	Retired	Semi-rural	IDP	7 months	No	No	3
P5	F	72	In couple	Entrepreneur	Retired	Urban	GP	2 years	Yes	No	1
P6	M	30	Single	Design engineer	Asset	Urban	GP	5 years	No	Yes	1
P7	F	56	Single	Healthcare aide	Asset	Rural	IDP	7 years	No	Yes	1
P8	M	68	In couple	Computer scientist	Retired	Urban	IDP	6 months	No	No	4
P9	F	44	Single	Fashion designer	Asset	Urban	IDP	2 years	Yes	No	3
P10	M	55	In couple	Teacher	Work stopping	Urban	IDP	6 months	No	Yes	1
P11	M	58	In couple	Police officer	Work stopping	Urban	IDP	9 months	No	No	2
P12	M	35	In couple	Electrician	Asset	Urban	IDP	5 years	Yes	No	3

F = Female; M = Male; IDP = Infectious disease physician; GP = General practitioner. * Clinical situation: 1 = confirmed diagnosis of LB, treated according to the recommendations, with persistent symptoms; 2 = diagnosis of LB disproved, patients had received or are receiving treatments contrary to recommendations, with persistent symptoms; 3 = patients undergoing diagnostic evaluation for unexplained symptoms, with no obvious diagnosis; 4 = confirmed diagnosis of LB, treated according to the recommendations, without persistent symptoms.

## Data Availability

The detailed transcriptions of the anonymised interviews in French can be requested at alice.raffetin@chiv.fr.

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
