# Peer review of "Perceptions, Representations, and Experiences of Patients Presenting Nonspecific Symptoms in the Context of Suspected Lyme Borreliosis"

_microorganisms, 2021, doi:10.3390/microorganisms9071515_

Round 1

Reviewer 1 Report

This is an interesting and well-written report.

Major remarks:

- The problem is whether 12 persons included in the study (of them 6 with a confirmed diagnosis of LB), really represent population of patients with (post LB) subjective symptoms. This drawback has been mentioned in discussion section but should have been more exposed.

- What were the criteria for confirmed LB and criteria for disproval of the diagnosis of LB?

- I agree that additional reference centers would make sense and be necessary, but there are many people with long-lasting symptoms that affect daily life (including those caused by LB or attributed to LB), so many additional centers would probably be needed. Another option would be that reference centres become a model for successful approaches, which would then be used in the wider environment.

Minor remarks:

A few typos.

Reviewer 2 Report

Summary
This paper reports the results of a study carried out with patients who had a suspicion of Lyme borreliosis. Interviews were carried out with 12 patients and a grounded theory approach was used for analyses. The results highlight the complex nature of patient experiences with this condition.

Review:

This is a strong paper presenting an interesting and important description of the experiences of individuals suffering from suspected LB. The results section, in particular, is well-written and provides clear themes with well-chosen supportive quotes.

I believe the following suggestions would strengthen the paper.

  1. The following papers are relevant to the topic here and could be added to the literature review (lines 66 – 76) and discussion: Drew & Hewitt (2006 – Public Health Nursing); Ali et al, (2014 – BMC Fam Pract); Rebman et al., 2015), Gaudet et al., 2019 – Healthcare; and especially Boudreau et al, (2018 – J. of Patient Experience).

  1. Regarding eligibility criteria (line 93): Please clarify whether the “suspicion of LB” is held by the doctor or the patient. In other words, did the sample include individuals who believed that they suffered from LB but did not have a physician who shared this suspicion? If not, then this selection bias should be discussed as a limitation to the study in the discussion, because it means that individuals who could not find any support at all from the medical community are not included in the sample.

  1. It is stated that the interviewee was a physician. Was this person involved in these patients care or linked to the clinic/hospital/sites where these patients were accessing care? Even if the interviewer was not involved in these specific patients’ care, the fact that the interviewer is a physician is a concern that should be addressed in the limitations section. Specifically, interviewees may have been less willing to made statements that were critical of MDs and the health-care system because they were talking to a physician.

  1. Please briefly explain the sentence “the interviewer’s status as a physician was carefully checked (line 121).

  1. In describing the sample, the authors mention “an association” line 145 and line 317. What is this association? In what way is it relevant to the study? Does this imply that the residents knew each other?

  1. The model presented on page 7 is quite complex (and possibly overly wordy) and more text could be provided to help the reader understand the figure.

  1. In my view, the discussion would read better if it were reorganized. Specifically, I think the limitations should be moved to the end (just before the final conclusion). Also, the sections (4.3, 4.4, 4.5) seem choppy and would work better if woven together into a narrative.

  1. In the discussion, Section 4.4 seems to reflect very much a physician’s perspective – and suggests (perhaps inadvertently) that well-informed patients are a problem. Is this what your data indicate? (On the contrary, the quotes provided seem to suggest that uninformed physicians were the problem?) I think that the authors could do a better job here of integrating/contrasting the data obtained in this study with past research representing physicians’ viewpoints.

  1. On line 326 it is stated that patients receiving care from the centres for tick-borne diseases were “satisfied” but no reference or data is provided to support this statement. How was it determined that they were satisfied? Is this anecdotal responses from patients? Based on interview responses?

  1. In the conclusion section, the authors state “the results of our study can be used to create a self-assessment questionnaire….” This phrasing is somewhat confusing: unless this questionnaire has already been designed and tested, perhaps “could be used” would be preferable phrasing. Better yet, the possibility of designing and testing this type of questionnaire could be presented as a possible new research directions on the same level as the study of patients receiving care from the new reference centres.

Reviewer 3 Report

My suggestions for authors are following:

  • Please take into notice that the France approach could not be applicable for many Lyme disease endemic countries,
  • Based on your study results please outline your practical recommendations for physicians,
  • Adjust your recommendations for the practical  needs of medical practitioners.

Please update my previous comments.

Round 2

Reviewer 1 Report

I think that the revised article is in good shape and warrants publication.    

Reviewer 3 Report

Thank you for the improvements, these are acceptable and change the article content more understandable